# Design of Paracetamol Delivery Systems Based on Functionalized Ordered Mesoporous Carbons

**DOI:** 10.3390/ma13184151

**Published:** 2020-09-18

**Authors:** Joanna Goscianska, Aleksander Ejsmont, Anna Olejnik, Dominika Ludowicz, Anna Stasiłowicz, Judyta Cielecka-Piontek

**Affiliations:** 1Department of Chemical Technology, Faculty of Chemistry, Adam Mickiewicz University in Poznań, Uniwersytetu Poznańskiego 8, 61-614 Poznań, Poland; aleejs@amu.edu.pl (A.E.); annamar@amu.edu.pl (A.O.); 2Department of Pharmacognosy, Faculty of Pharmacy, Poznań University of Medical Sciences, Święcickiego 4, 61-781 Poznań, Poland; dominika.siakowska@interia.eu (D.L.); astasilowicz@ump.edu.pl (A.S.)

**Keywords:** carbon carriers, active pharmaceutical ingredients, adsorption of paracetamol, kinetic modelling, permeability study

## Abstract

The oxidized ordered mesoporous carbons of cubic and hexagonal structure obtained by two templating methods (soft and hard) were applied for the first time as delivery systems for paracetamol—the most common antipyretic and analgesic drug in the world. The process of carbon oxidation was performed using an acidic ammonium persulfate solution at 60 °C for 6 h. The functionalization was found to reduce the specific surface area and pore volume of carbon materials, but it also led to an increasing number of acidic oxygen-containing functional groups. The most important element and the novelty of the presented study was the evaluation of adsorption and release ability of carbon carriers towards paracetamol. It was revealed that the sorption capacity and the drug release rate were mainly affected by the materials’ textural parameters and the total amount of surface functional groups, notably different in pristine and oxidized samples. The adsorption of paracetamol on the surface of ordered mesoporous carbons occurred according to different mechanisms: donor–acceptor complexes and hydrogen bond formation. The adsorption kinetics was assessed using pseudo-first- and pseudo-second-order models. The regression results indicated that the adsorption kinetics was more accurately represented by the pseudo-second-order model. Paracetamol was adsorbed onto the carbon materials studied following the Langmuir type isotherm. The presence of oxygen-containing functional groups on the surface of ordered mesoporous carbons enhanced the amount of paracetamol adsorbed and its release rate. The optimal drug loading capacity and expected release pattern exhibited oxidized ordered mesoporous carbon with a hexagonal structure obtained by the hard template method.

## 1. Introduction

The ever-increasing cost of the development of new therapeutic drugs, the long time it takes for their introduction onto the market, and the high risk of failure in clinical trials have stimulated efforts directed to the design of delivery systems of already known active pharmaceutical ingredients (APIs) with well-defined safety profiles and physicochemical properties [1]. Therefore, a lot of attention in the latest years has been given to develop and produce stable and selective API carriers. The use of an appropriate carrier and suitable method of active pharmaceutical ingredient incorporation into its structure can bring an improvement in the API bioavailability, prevent its recrystallization, limit the side effects, and lengthen the activity time in the human body [1,2,3]. Miscellaneous drug carriers have been developed and investigated, each of which has distinctive advantages. The most popular types of pharmaceutical vehicles include liposomes, lipid and metal nanoparticles, carbon nanotubes, mesoporous carbons, metal–organic frameworks (MOFs), and mesoporous silica of different structures such as MCM-41, MCM-48, SBA-15, and SBA-16 [1,2,3,4,5,6,7,8,9,10]. Amidst them, especially highly biocompatible ordered mesoporous carbons (OMCs) containing pores with a size ranging from 2 to 50 nm, which form a two- or three-dimensional network, can precisely tune the drug release rate and thus prolong its therapeutic effect [7,8]. Their peculiar features can be achieved in two main synthetic procedures, including hard template [11,12,13] (also called nanocasting) or soft template methods [14]. In the nanocasting, the ordered mesoporous silica (OMS) or zeolites are used as scaffolds. They are initially impregnated with precursors rich in carbon (e.g., sugars, furfuryl alcohol) and then carbonized at high temperature. In the last step, the matrices are removed by strong base or hydrofluoric acid [11,12,13]. Contrarily, in the soft template procedure, the OMCs are synthesized by co-assembly of carbon precursors (phenolic resins) and triblock copolymers (organic surfactants), which are subjected to carbonization [14]. After carbonization, the ordered mesoporous carbons are obtained that, depending on the selected method, are different not only in structure, but also in pores’ ordering, morphology, and uniformity. Such materials are featured by well-developed specific surface area, large pore volume, and very good mechanical, thermal, and chemical stability. It allows for a wide range of post-synthetic modifications/functionalizations to proceed, for example, via grafting [15], impregnation [16], and sulfonation [17], but mostly through diverse oxidation [18,19,20]. Different oxidizing agents at varied temperatures enrich OMCs’ surface in additional functional groups [21]. During such processes, it is important to preserve the material ordered structure and enhance their sorption capabilities towards the required adsorbate. Active pharmaceutical ingredients are especially challenging adsorbate because their bioavailability depends on many aspects that the drug carrier is supposed to maintain. Pharmaceutics usually exhibit low solubility, chemical lability, or poor permeability through biological membranes [22]. Therefore, the carriers applied have to be stable in aqueous media, harmless in biological systems, and should have a good affinity towards API. Moreover, the drug should also have the ability to release from the vehicle in required environment in the controlled manner [23,24,25,26]. In this study, paracetamol (also known as acetaminophen) was used as a model drug. This medicine is commonly applied to treat fever, cold and flu symptoms (when combined with decongestants and antihistamines), and mild and moderate pain [27]. Paracetamol is also recommended to control the fever symptoms of patients with coronavirus disease 2019 (COVID-19), who require conservative management and palliative care [28,29]. The overdosage of paracetamol may be responsible for liver damage, resulting in hepatic necroses. Acetaminophen is available in different forms, including capsules, tablets, liquids, soluble powders, and suppositories [27]. Furthermore, paracetamol can be delivered into the systemic blood from modified or immediate-release formulations. In order to control its dissolution over a certain time, it is essential to develop an appropriate vehicle that will improve drug efficiency. Therefore, the aim of the presented research was to synthesize pristine and oxidized ordered mesoporous carbons (ox-OMCs) of a different structure by the hard and soft template method and apply them as a novel platform for paracetamol. It is worth noting that, by tuning the type of carbon structure and introducing oxygen functionalities, it is possible to modulate the course of the drug release processes. Paracetamol has a relatively short half-life (4 h) and low solubility; therefore, the development of a carrier that will increase drug solubility and prolong its half-life was the target of this study. The effort was devoted to developing new nanomaterial-based therapeutics that will provide better drug release control.

## 2. Materials and Methods

### 2.1. Preparation of Mesoporous Carbon Carriers via Hard Template Method

The hard template method was used to synthesize ordered mesoporous carbons C_KIT-6_ (cubic structure) and C_SBA-15_ (hexagonal structure). In the first stage, OMS matrices KIT-6 and SBA-15 were prepared according to the reported procedures [5,6].

In the hydrothermal synthesis of KIT-6, the triblock copolymer Pluronic P123 (4 g, EO_20_PO_70_EO_20,_ Aldrich, St. Louis, MO, USA) was dissolved in the acidic solution (144 g of distilled water and 7.9 g of hydrochloric acid, Avantor Performance Materials Poland S.A.) at 35 °C. Subsequently, butan-1-ol (4 g, POCh) and tetraethyl orthosilicate (8.6 g, 98% wt, Aldrich) were added. The prepared solution was intensively mixed for 24 h at 35 °C and subjected to hydrothermal treatment for 24 h at 100 °C. The received precipitate was filtered, washed three times with distilled water, and dried in the oven at 100 °C overnight. The removal of triblock copolymer proceeded through calcination at 550 °C for 8 h.

The substrates for SBA-15 preparation were triblock copolymer Pluronic P123 (0.5 g, Aldrich), hydrochloric acid (19 mL, 1.6 mol/L, Avantor Performance Materials Poland S.A., Gleiwitz, Poland), and TEOS (1.1 mL, tetraethyl orthosilicate, 98% wt., Aldrich). To an aqueous hydrochloric acid solution of Pluronic P123 maintained at 35 °C, TEOS was added dropwise upon stirring continuously for 6 h. Then, the as-prepared mixture was subjected to hydrothermal treatment in tightly closed polypropylene bottles in an oven for 24 h at 35 °C as a first stage, and for 6 h at 100 °C as a second stage. Then, the material obtained was filtered, washed three times, and dried at 100 °C for 12 h. Lastly, to remove the template, it was calcined for 8 h at 550 °C.

KIT-6 and SBA-15 silica materials were subjected to twice repeated impregnation with a sucrose solution. An exactly weighted portion of sucrose (1.25 g, Aldrich) was dissolved in sulfuric(VI) acid (0.14 mL, Avantor Performance Materials Poland S.A.) and distilled water (5 mL). Next, as-prepared solution was added slowly to the flask containing OMS. The contents were heated in the oven firstly for 6 h at 100 °C, and then for 6 h at 160 °C. Afterwards, the obtained silica–carbon composites were treated again with a solution containing sucrose (0.8 g), sulfuric(VI) acid (0.09 mL), and distilled water (5 mL). The materials were heated in the oven for 6 h at 100 °C and then for the next 6 h at 160 °C. The composites obtained were carbonized for 3 h at 900 °C, at the temperature increase rate of 2.5 °C/min, and the remaining silica was washed out twice with 200 mL 5% of hydrofluoric acid solution (Avantor Performance Materials Poland S.A.). The materials were collected by filtration, washed with ethanol three times, and dried for 12 h at 100 °C. The carbon materials were labelled as C_KIT-6_ and C_SBA-15_, respectively.

### 2.2. Preparation of Mesoporous Carbon Carriers via Soft Template Method

Mesoporous carbon C_ST_ was obtained via the soft template method based on the co-assembly triblock copolymer Pluronic F127 (Sigma-Aldrich) and carbon precursor—resorcinol (Sigma-Aldrich). In the initial stage, Pluronic F127 (1.875 g) and resorcinol (1.88 g) were dissolved in the solution of ethanol (POCh, 96%) and distilled water (15.38 g, weight ratio C_2_H_5_OH/H_2_O = 10:7) at room temperature and the as-prepared mixture was stirred vigorously. Subsequently, hydrochloric acid (0.14 mL, POCh, 36%) and formaldehyde solution (1.93 mL, Chempur, 37%) were added. The solution was stirred intensively until it became turbid. Two hours later, the two-phased mixture was separated; one (aqueous layer) was removed, while the other (organic layer) was stirred by a magnetic mixer for 72 h. The dark brown monolith received at this stage was subsequently heated up to 100 °C and kept for 24 h in a propylene bottle. Finally, it was carbonized in a tube furnace under nitrogen atmosphere at three stages: 5 h—180 °C, 4 h—400 °C, and 2 h—800 °C.

### 2.3. Functionalization of Mesoporous Carbon Carriers

OMC carriers were subjected to oxidation using acidic ammonium persulfate solution (APS, Sigma-Aldrich) with a concentration of 1 mol/L as a gentle oxidant. This procedure was applied to generate oxygen functionalities on the carbon surface. In a round-bottomed flask, the carbon materials (0.5 g) were flooded with APS solution (30 mL). The process of oxidation was performed under reflux upon vigorous stirring at 60 °C. After 6 h, the solids were filtered off, washed with ethanol and distilled water, followed by drying at 100 °C overnight. The oxidized carbon samples were denoted as C_SBA-15_-APS, C_KIT-6_-APS, and C_ST_-APS.

### 2.4. Characterization of Materials

#### 2.4.1. Low-Temperature Nitrogen Sorption

The pore structure of the synthesized carbon carriers was characterized by low-temperature nitrogen adsorption/desorption isotherms measured at −196 °C with the use of a Quantachrome Autosorb IQ apparatus. Before adsorption measurements, the pristine carbon samples were degassed in vacuum at 300 °C for 3 h, while oxidized carbon materials were degassed in vacuum at 150 °C for 3 h. The Brunauer–Emmett–Teller (BET) method was utilized for the determination of the surface areas (*S_BET_*) of carbon carriers. The average pore size was estimated from the adsorption branch of isotherm using the Barret–Joyner–Halenda (BJH) method.

#### 2.4.2. Powder X-ray Diffraction

The type and ordering of the mesoporous structure of the carbon carriers were identified by powder X-ray diffraction (XRD). XRD patterns were made at room temperature with a step size 0.02° in the small-angle range using a D8 Advance Diffractometer (Bruker) with the copper Kα1 radiation (λ = 1.5406 Å).

#### 2.4.3. Surface Oxygen Functional Groups

The number of surface oxygen functional groups of acidic and basic nature was determined by the Boehm method [30]. In the case of acidic groups, mesoporous material (0.2 g) was suspended in sodium hydroxide solution (25 mL, 0.1 mol/L, Chempur, Karlsruhe, Germany) and agitated at room temperature for 24 h. Afterwards, the liquid was separated from the solid sample by centrifugation for 10 min and titrated with a hydrochloric acid solution (0.1 mol/L, Chempur) in the presence of methyl orange as an indicator. In order to establish the total content of basic oxygen groups, the converse procedure was applied.

#### 2.4.4. Infrared Spectroscopy

FT-IR (Fourier-transform infrared) spectra of the carbon materials before and after paracetamol adsorption were registered with the use of a Varian 640-IR spectrometer. The samples were studied in the form of tablets, obtained by pressing a mixture of anhydrous KBr (ca. 0.25 g) and the carbon material (0.3 mg) in a special steel ring, under a pressure of 10 MPa. The analysis was carried out in a wavenumber range of 4000–400 cm^−1^ (at a resolution of 0.5 cm^−1^; number of scans: 64).

### 2.5. Paracetamol Adsorption Studies

In order to evaluate the adsorption abilities of the carbon materials towards paracetamol (PAR), a series of its solution was prepared, the concentration of which varied from 5 to 150 mg/L. The samples (0.025 g) were placed in flasks and flooded with 50 mL of a paracetamol solution of a certain concentration, and the contents were shaken in the temperature-controlled orbital shaker (KS 4000i control, IKA, Staufen im Breisgau, Germany) at a fixed shaking rate of 250 rpm over 24 h. After that, the drug solutions were separated from the adsorbents by centrifugation for 10 min and their absorbance was studied with the use of Agilent Cary 60 UV/vis spectrophotometer at the wavelength of 243 nm.

The amount of the paracetamol adsorbed per unit weight of OMCs, *q_e_* (mg/g), was calculated according to the following equation:(1)$$q_{e}=\frac{(C_{0}-C_{e})\cdot V}{m}$$
where *C*_0_ is the initial concentration of paracetamol (mg/L), *C_e_* is the residual concentration of paracetamol (mg/L), *V* is the volume of the paracetamol solution (L), and *m* is the mass of the carrier (g).

Analysis of the adsorption data was carried out using Freundlich and Langmuir models [31,32]. The criterion of best fitting is the correlation coefficient R^2^. The Langmuir isotherm is described by the following linear equation [31]:(2)$$\frac{C_{e}}{q_{e}}=\frac{1}{q_{m}K_{L}}+\frac{C_{e}}{q_{m}}$$
where *C_e_* is the equilibrium concentration of paracetamol (mg/L), *q_e_* is the quantity of drug adsorbed onto the adsorbent at equilibrium (mg/g), *q_m_* is the maximum monolayer adsorption capacity of adsorbent (mg/g), and *K_L_* is the Langmuir constant denoting the energy of adsorption and affinity of the binding sites (L/mg).

The linear form of the Freundlich equation is as follows [32]: (3)$$\ln q_{e}=\ln K_{F}+\;\frac{1}{n}\ln C_{e}$$
where *q_e_* is the amount of paracetamol adsorbed at equilibrium (mg/g) and *C_e_* is the equilibrium concentration of the drug (mg/L). *K_F_* and *n* are the Freundlich constants; *n* gives an indication of how favorable the adsorption process is and *K_F_* (mg/g (L/mg)^1/n^) is related to the adsorption capacity of the adsorbents.

The kinetic studies of paracetamol adsorption were carried out to understand the adsorption rate and mechanism at the solid–liquid interface of mesoporous carbon carriers and drug molecules. In this context, linear forms of the pseudo-first- and pseudo-second-order models were applied to estimate the adsorption process by fitting the experimental data obtained. These models are given in Equations (4) and (5), in the same order [33,34]:(4)$$\ln\left(q_{e}-q_{t}\right)=\;\ln\;q_{e\;}-\;\frac{k_{1}t}{2.303}$$
(5)$$\frac{t}{q_{t}}=\;\frac{1}{k_{2}q_{e}^{2}}+\;\frac{t}{q_{e}}\;$$
where *q_e_* is the amount of the paracetamol adsorbed at equilibrium state (mg/g), *q_t_* is the amount of the paracetamol adsorbed in time (mg/g), *k*_1_ is the rate constant of adsorption in the pseudo-first-order model (min^−1^), and *k*_2_ is the rate constant of adsorption in the pseudo-second-order model (g/mg min).

### 2.6. Paracetamol Release Studies 

Pure paracetamol and ordered mesoporous carbons with the adsorbed paracetamol were weighed to gelatin capsules placed in the springs in order to sink and prevent flotation on the surface of the medium. The analyses were performed using USP (United States Pharmacopoeia) dissolution paddle apparatus (Agilent 708-DS) in the gastric juice medium (pH 1.2) maintained at 37 °C and stirred at 50 rpm. At the defined time intervals, 5.0 mL of dissolution samples was withdrawn and replaced with an equal volume of temperature-equilibrated medium, and then filtered through a 0.45 μm membrane filter. The changes in concentration of paracetamol were measured using high-performance liquid chromatography (HPLC) with a DAD (Diode Array Detector) detector. The separations were performed using a stationary phase based on a Kinetex-C18 column (100 mm × 2.1 mm; 5 µm) at 37 °C. The mobile phase consisted of 0.1% formic acid and acetonitrile (90:10, *v/v*) with the flow rate of 0.5 mL/min. The injection volume was 5 µL and the detection wavelength was set at 243 nm.

The paracetamol release data were fitted to kinetic models including zero-order (percentage of acetaminophen release versus time), first-order (log of the percentage of acetaminophen remaining versus time), Higuchi’s model (percentage of acetaminophen release versus square root of time), the Korsmeyer–Peppas model (log of the percentage of acetaminophen release versus log time), and the Hixson–Crowell model (cube root of the percentage of acetaminophen remaining versus time) [35,36]. Additionally, R^2^ was calculated to determine which model follows the selected release profile.

The two-factor values *f*_1_ and *f*_2_ introduced by Moore and Flanner were used to compare dissolution profiles. The *f*_1_ and *f*_2_ values are defined by the following equations:(6)$$f_{1}=\frac{\sum_{j=1}^{n}\left|R_{j}-T_{j}\right|}{\sum_{j=1}^{n}R_{j}}\times 100$$
(7)$$f_{2}=50\;\times \log\left({\left(1+\left(\frac{1}{n}\right)\overset{n}{\underset{j=1}{\sum }}{\left|R_{j}-T_{j}\right|}^{2}\right)}^{-\frac{1}{2}}\times 100\right)$$
where *n* is the number of time points, *R_j_* is the percentage of the reference dissolved product in the medium, *T_j_* is the percentage of the dissolved tested product, and *t* is the time point. Dissolution profiles are similar when the *f*_1_ value is close to 0 and *f*_2_ is close to 100 (between 50 and 100); Table 1 [37].

### 2.7. Permeability Study

In vitro gastrointestinal (GIT) permeability test was performed using PAMPA (parallel artificial membrane permeability assay). The kit consists of a 96-well microfilter plate divided into two chambers, donor and acceptor, separated by a 120 μm thick microfilter disc coated with a 20% (*w*/*v*) dodecane solution of a lecithin mixture (Pion, Inc., Billerica, MA, USA). The samples solutions were added to the donor compartments. After adding the acceptor solution to the acceptor wells, both parts—donor and acceptor—were placed together, and the sandwich was incubated for 3 h at the temperature of 37 °C in a humidity-saturated atmosphere. Afterwards, both chambers were split, and the concentrations of donor and acceptor solutions were measured using UV spectroscopy at 243 nm. The apparent permeability coefficient (*P_app_*) was calculated using the following equation:(8)$$P_{app}=\frac{-ln\left(1-\frac{C_{A}}{C_{equilibrium}}\right)}{S\times \left(\frac{1}{V_{D}}+\frac{1}{V_{A}}\right)\times t}$$
where *V_D_* is the donor volume; *V_A_* is the acceptor volume; and *C_equilibrium_* is the equilibrium concentration, where $C_{equilibrium}=\frac{C_{D}\times V_{D}+\;C_{A}\times V_{A}}{V_{D}+V_{A}}$, *S* is the membrane area and *t* is the incubation time (in seconds). Compounds that have a *P_app_* < 0.1 × 10^−6^ cm/s are referred as ones with low permeability, compounds found to have medium permeability have a 0.1 × 10^−6^ cm/s ≤ *P_app_* < 1 × 10^−6^ cm/s, and compounds with a *P_app_* ≥ 1 × 10^−6^ cm/s are classified as ones with high permeability [38].

## 3. Results and Discussion

### 3.1. Physicochemical Characterization of Mesoporous Carbon Carriers

Small-angle XRD patterns of OMCs synthesized by hard and soft template methods before and after oxidation with APS are displayed in Figure 1. The diffractogram of the pristine C_KIT-6_ sample shows a strong peak at 2Θ ≈ 1° and less intensive reflections in the range 2Θ ≈ 1.5–2.3°, indicating the presence of an ordered cubic structure with Ia3d symmetry (Figure 1a). On the other hand, in the case of the XRD profile of the carbon C_SBA-15_, an intensive peak at 2Θ ≈ 1° characteristic for hexagonal pore arrangement is noted. Moreover, the reflections at 2Θ ≈ 1.7–2.5°, corresponding to the planes (100), (110), and (200) of p6mm structure, are also observed (Figure 1b). It was established that the use of a gentle oxidant, ammonium persulfate, for functionalization of materials, does not significantly affect the ordering of the cubic and hexagonal mesoporous structure of C_KIT-6_ and C_SBA-15_ samples, respectively. Pristine and oxidized carbons obtained by soft templating are characterized by a less ordered mesoporous structure, as evidenced by a low-intensity peak at the angle 2Θ ≈ between 0.5 and 1° (Figure 1c). The expected ordering of the 2D mesoporous structure of C_ST_ and C_ST_-APS is hexagonal; however, it is not possible to determine the symmetry group from the XRD data.

The textural features of nanomaterials have a major impact on the mass transport, accessibility of adsorption sites for different active pharmaceutical ingredients, and their adsorption capacity. The data on the specific surface area, pore volume, and size of all mesoporous carbon carriers are collected in Table 2. According to these results, the synthesis of ordered mesoporous carbons C_KIT-6_ and C_SBA-15_ on the basis of silica matrices KIT-6 and SBA-15, respectively, provides materials with a well-developed BET surface area (S_CKIT-6_—1003 m^2^/g, S_CSBA-15_—986 m^2^/g) and total pore volume (V_CKIT-6_—1.15 cm^3^/g, V_CSBA-15_—1.47 cm^3^/g). The carbon C_ST_ obtained by the soft templating is characterized by a much smaller specific surface area (526 m^2^/g) and pore volume (0.49 cm^3^/g). It should be noted that, regardless of the synthesis method used, all materials contain micropores in the structure whose surface area is 306 m^2^/g for C_KIT-6_, 545 m^2^/g for C_SBA-15_, and 231 m^2^/g for C_ST_. They are probably located within the walls of the mesopores. As follows from Table 2, although the mesostructural regularity of the carbonaceous carriers is preserved, after oxidation with APS, their textural parameters deteriorate considerably in comparison with the non-functionalized materials. It is assumed that the modification process with a gentle oxidation agent takes place primarily inside micropore/small mesopore, which may be due to their high potential to easily attach oxygen-containing functional groups. Consequently, the oxygen-containing groups can partially block the pores in the structure of carbon materials, leading to decreasing their volume and surface area.

The surface chemistry of materials has a direct impact on their sorption capacities towards active pharmaceutical ingredients. Figure 2 depicts the content of oxygen functional groups of an acidic and basic nature on the surface of mesoporous carbon carriers determined using the Boehm method [30]. The oxidation of C_KIT-6_, C_SBA-15_, and C_ST_ samples with APS causes a significant increase in the amount of acidic groups, which are favorable especially for the adsorption of guest molecules from polar solvents. Interestingly, the total number of acidic groups on the surface of C_KIT-6_-APS (4.03 mmol/g) and C_SBA-15_-APS (4.00 mmol/g) materials is similar. The oxidized carbon C_ST_-APS synthesized by the soft template method contains a lower number of acidic groups (2.11 mmol/g) compared with other oxidized samples. Moreover, it was established that C_SBA-15_, C_KIT-6_-APS, and C_ST_-APS samples do not possess oxygen groups of a basic nature on the surface. In the case of the C_SBA-15_-APS carbon sample, during the oxidation process with APS solution, a small amount of chromene and pyrone-like groups was generated on its surface. Therefore, an increase in the content of basic functional groups was observed.

### 3.2. Paracetamol Adsorption and Release Studies

For the purpose of setting the time required for reaching equilibrium between the paracetamol molecules and OMCs, the kinetics of adsorption was thoroughly analyzed. As presented in Figure 3, the uptake of the drug molecules was very fast in the first 10 min of the process. This demonstrates that, on the surface of carbonaceous carriers, a large number of vacant adsorption sites occurred, and paracetamol could be adsorbed with ease. With the extending contact time, drug molecules penetrated further and deeper within the pores. They also came across a greater resistance if the process was continued. The adsorption process slowed down considerably. After 60 min, no increase in the amount of adsorbed paracetamol was noted, which means that a state of equilibrium was reached and the limited number of active sites on the OMC samples’ surface were engaged. In this study, the experimental data were fitted to the pseudo-first-order kinetic model of Lagergren and pseudo-second-order kinetic model of Ho and McKay [33,34]. The values of *k*_1_ and *k*_2_ constants, correlation coefficients (R^2^), and the theoretical amounts of the paracetamol adsorbed (*q_e(cal)_*) on the surface of carbon materials are collected in Table 3. The *k*_1_ constants were estimated from the plots of ln(*q_e_−q_t_*) versus *t*, while those of *k*_2_ were estimated from the plots of *t/q_t_* versus *t*. The low correlation coefficients for the pseudo-first-order model (R^2^ = 0.927–0.987; Table 3) exclude the possibility of its application to describe the mechanism of paracetamol adsorption onto mesoporous carbon samples. Moreover, the *q_e(cal)_* values calculated on the basis of the linear plots are significantly lower than those corresponding experimental *q_e(exp)_* values. The experimental data revealed better consent with the pseudo-second-order kinetic model, suggested by higher correlation coefficient values (R^2^ = 0.999; Table 3). It signifies that this kinetic model can be used to predict the amount of drug adsorbed at different contact time intervals by OMCs obtained via hard and soft template methods. It should be mentioned that, according to this model, chemisorption takes place in addition to physisorption. These processes depend on the properties of both the adsorbents and the adsorbates.

Figure 4 depicts the equilibrium adsorption isotherms of paracetamol onto pristine and ox-OMCs in aqueous solution. It was observed that the amount of paracetamol adsorbed significantly increases with an increasing initial concentration of its solutions until the adsorption reaches a saturation point. This may be due to the occurrence of the dynamic interplay between the adsorbate and the carbon adsorbents taking place on active sites characterized by a progressive affinity for the drug species. Among the pristine mesoporous carbons, C_KIT-6_ and C_SBA-15_ exhibited higher sorption capacity towards paracetamol than the C_ST_ sample, which is related to their better developed specific surface area and larger pore volume. The type of mesoporous structure does not considerably affect the drug adsorption process. According to the results, the oxidation of carbon materials with the use of APS leads to an increase in their sorption capacity towards paracetamol. Similar results were observed by Liang et al. [39], who detected that, when activated carbon had an acidic character and contained a high concentration of oxygen groups on the surface, more acetaminophen could be loaded into these materials. Therefore, probably the most important factor determining the amount of the adsorbed drug is the content of the oxygen functionalities on the OMC surfaces. During the adsorption of paracetamol, acceptor–donor complexes can be formed between the groups containing free electron pairs (e.g., oxygen in phenolic/carboxylic groups) that are present on the surface of OMCs and the electropositive nitrogen in paracetamol molecules. The paracetamol exhibits proton acceptor and donor groups. However, the resonance-generated changes result in nitrogen passing from the proton acceptor to the donor [40]. Therefore, hydrogen bond interactions can also occur during the drug adsorption process. They will be more intense in mesoporous carbons possessing a higher concentration of proton acceptor functional groups.

At the equilibrium of the adsorption process, the adsorption isotherm can be used to estimate the distribution of adsorbate molecules between the liquid and solid phases. In this work, analysis of the experimental results was performed with the use of the Langmuir and Freundlich adsorption models [31,32]. All parameters—*q_m_, K_L_,* 1*/n,* and *K_F_*—and correlation coefficients R^2^ are listed in Table 4. They were computed from the intercept and linear gradient of the graphs of *C_e_/q_e_* and *C_e_* for the Langmuir isotherm (Figure 5a) and from the plots of *ln(q_e_)* against *ln(C_e_)* for the Freundlich model (Figure 5b). On the basis of the values of the R^2^ correlation coefficient (0.999), it was established that the Langmuir isotherm appropriately describes the results obtained for the adsorption of paracetamol on the surface of pristine and oxidized mesoporous carbon materials. The experimental data revealed that the maximum sorption capacity (*q_e_*) of all synthesized samples towards drug is slightly lower from those estimated theoretically (*q_m_*). On the basis of these results, it can be stated that paracetamol molecules were adsorbed on the surface of carbon materials by forming a homogenous monolayer. The factor *1/n* computed from the Freundlich isotherm was lower than 1 for all carbon adsorbents, meaning that adsorption of the drug was energetically favorable and straightforward to perform (Table 4).

The effectiveness of the paracetamol adsorption process on the surface of ordered mesoporous carbon materials was also studied by infrared spectroscopy (Figure 6). The FT-IR spectra of pristine and functionalized OMCs were discussed in our previous paper [41]. They clearly demonstrate that the process of oxidation with an acidic solution of ammonium persulfate leads to the generation of a high density of carboxylic, ketone, phenolic, and etheric groups (Figure 6a). After drug loading, the significant differences in the FT-IR spectra of nanomaterials were detected. The absorption bands at wavenumbers around 3100–3700 cm^−1^ can be assigned to overlapped N-H and O-H stretching vibrations present in the adsorbed molecules of paracetamol. The C=O stretching vibrations were observed at 1660 cm^−1^, while C-H bending vibration was detected at 1400 cm^−1^. Moreover, C_Ar_-N stretching vibrations were identified at 1232 cm^−1^. Additionally, the vibrations of the aromatic ring (C_Ar_-C_Ar_) were found at 1600–1500 cm^−1^ [42,43,44]. The absorption bands at 2300 cm^−1^ corresponded to N-H/C-O stretching vibrations that appeared as a result of the interactions between paracetamol and mesoporous carbon materials (Figure 6b).

The paracetamol release studies were performed in simulated gastric fluid (pH 1.2). Primarily, the drug release profiles from gelatin capsule and ordered mesoporous carbon systems were compared. Figure 7 presents the dissolution profile of pure paracetamol from a gelatin capsule. After 10 min of the analysis, ca. 100% of acetaminophen was detected in the medium, so immediate drug release was observed. On the other hand, the sustained release pattern was identified when paracetamol was diffused from non-functionalized ordered mesoporous carbon materials (Figure 8). The quantity of drug released within 2 h diminished in the following sequence C_ST_ > C_KIT-6_ > C_SBA-15_. Paracetamol molecules were mostly loaded inside the pores of materials during the adsorption process and could not be freely liberated to acceptor medium. This phenomenon was most noticeable for C_SBA-15_, in which case ca. 28% of drug was released within 120 min. This sample showed the largest pore diameter (6.54 nm) compared with other materials C_KIT-6_ (5.78 nm) and C_ST_ (4.12 nm) (Table 2). On the basis of these parameters, it could be stated that the amount of paracetamol released decreased as the pore diameter increased. It should be highlighted that C_SBA-15_ also exhibited the largest micropore surface area and the highest micropore volume, which allowed the drug molecules to be loaded inside, and as a consequence, they were not easily desorbed. These results proved that the textural parameters and the structure of carbon materials influenced the percentage of acetaminophen that was diffused to acceptor medium. Similar conclusions were reached when the paracetamol was released from activated and pristine carbon powders. The properties of these materials such as mesopore and micropore volume sizes determined the drug release pattern [45]. These observations were consistent with other studies in which porous materials such as zirconia/silica hybrids were applied as paracetamol carriers. The data obtained in this study proved that the amount of drug diffused to receptor medium was determined by chemical compositions of these materials (ZrO_2_-SiO_2_) and their textural parameters [46]. Moreover, it was established that the porosity of nanocarrier and the quantity of functional groups on its surface had an influence on the drug release ability.

Therefore, the paracetamol was also released from functionalized mesoporous carbon vehicles. In order to compare the data obtained, the dissolution profiles of paracetamol from pristine and oxidized OMCs are presented in Figure 9A–C. The results proved that all functionalized materials showed a higher percentage of drug release than the pristine samples. Owing to the oxidation of ordered mesoporous carbons, the paracetamol molecules were gathered mostly on the external surface of these materials, thus they could be easily desorbed from the modified materials.

The most significant difference in the amount of acetaminophen released was observed between C_SBA-15_ and C_SBA-15_-APS. For C_SBA-15_-APS, the initial burst release of around 73% was detected in the first 10 min of the analysis, and afterwards, the amount of drug increased steadily to reach ca. 92%. Meanwhile, for C_KIT-6_-APS in the beginning of the analysis, ca. 40% of paracetamol was detected in the acceptor fluid to achieve 67% after 120 min. The difference between the drug release from modified and non-modified C_KIT-6_ is mainly because of various acid–base properties of both materials. Therefore, in this case, the textural parameters are not as important as the number and types of groups attached to the surface of the material. The total number of acidic groups on the surface of C_KIT-6_-APS was almost two times higher compared with C_KIT-6_. It could be suggested that, for carbon material decorated with functional groups, host–guest interactions were different compared with that of pristine material, which had influence on the drug release pattern. A much lower difference in the percentage of drug released between the pristine and functionalized sample was observed for C_ST_ and C_ST_-APS samples. This could be justified by a similar total content of oxygen functional groups on the surface of the pristine material (1.85 mmol/g) and functionalized one (2.11 mmol/g). Two distinctive release steps were also observed when metal–organic frameworks (MIL-53(Fe) congruous and MIL-101) were applied as carriers for paracetamol. The drug was released slowly in a diffusion-controlled manner from MIL-53 (Fe) in 6 days. Because of the larger pore diameter and poorer host–guest interactions, the release of paracetamol from MIL-101 was faster than from MIL-53(Fe). However, the quickest diffusion of acetaminophen (in less than one hour) was observed from SBA-15 [47]. This phenomenon was associated with the large pore diameter of mesopores and competition between the paracetamol molecules and water during the adsorption process [48]. It was proved that drug release is dependent not only on its diffusion from the pores of materials, but also on host–guest interactions that occurred between the carrier and active compound. Mesoporous silica nanomaterials were also suggested as vehicles that could deliver the drug in a controlled, sustained pattern. Paracetamol was added at the beginning of the synthesis of these materials to bind drug to the silica network by van der Waals interactions. The drug release kinetics proceeded in two release stages, a fast release observed in the first hours and then sustained release. The amount of paracetamol diffused from SiO_2_ was 60% after 3 h and 80% after 200 h [49]. On the other hand, only 27% of paracetamol in 12 h was released from activated carbon powder, the drug loading capacity of which was 281 mg/g [39]. In another study, activated carbon was also applied as a vehicle for paracetamol [50]. The complete drug release was observed after 10 min of the analysis (1% of SDS, sodium dodecyl sulfate was added to the buffer solution). However, the experiment was carried out only in buffer medium at pH 5.8 and pH 7.2. There was no analysis performed in simulated gastric fluid.

In order to understand the paracetamol diffusion mechanism from carbon nanocarriers, the results were fitted to five different kinetic models that are the most commonly applied in drug release studies (Table 5). The results proved that, for all pristine carbon samples, the highest values of R^2^ were detected for the Higuchi model. Therefore, it can be assumed that paracetamol release was controlled by diffusion for C_SBA-15_, C_KIT-6_, and C_ST_. Meanwhile, for oxidized materials, the drug release was the most consistent with the Korsmeyer–Peppas model. On the basis of the release exponent (n value), it is possible to characterize which type of diffusion follows the drug (n < 0.45 corresponds to Fickian diffusion, 0.45 < n < 0.89 is related to non-Fickian diffusion, n = 0.89 corresponds to case II transport—zero order release, and n > 0.89 is associated with super case II transport) [51]. For C_SBA-15_-APS and C_ST_-APS, the paracetamol release was driven by Fickian diffusion as a result of chemical potential gradient. Meanwhile, for C_KIT-6_-APS, the n value was higher than 0.45, which indicated the non-Fickian diffusion mechanism.

The behavior of paracetamol release from ordered mesoporous carbons was compared with the dissolution profiles of pure paracetamol by the determination of *f*_1_ and *f*_2_ factors (Table 6). The release profiles are considered to be similar when *f*_1_ (difference factor) is close to 0 (range 0–15) and *f*_2_ (similarity factor) is close to 100 (range 50–100). The *f*_1_ and *f*_2_ factors determined for paracetamol in combination with the selected ordered mesoporous carbon carriers indicate that the release profiles obtained differ from the dissolution profiles of pure paracetamol. It proved that the drug was released in a modified pattern.

Additionally, the parallel artificial membrane permeability assay was applied as an in vitro model of passive transcellular transport of the active compound (Figure 10). According to the literature, the results obtained correlate well with in vivo drug absorption [52]. For all system based on oxidized mesoporous carbon materials, the values of *P_app_* were higher than 1.0 × 10^−6^ cm/s, which classified them as highly permeable.

## 4. Conclusions

In this study, new delivery systems for paracetamol were designed. They were based on non-toxic ordered mesoporous carbons of cubic and hexagonal structure obtained via hard and soft template methods. The carbon materials were oxidized with an acidic solution of APS at 60 °C for 6 h. The functionalization brought about a considerable depletion of the specific surface area and pore volume of materials, but it concomitantly led to the generation of acidic oxygen-containing functionalities. It was suggested that functional groups are attached primarily inside micropores/small mesopores, which are partially blocked. The OMCs modified in the previously mentioned manner turned out to be a very efficient adsorbents of paracetamol from aqueous solutions. The adsorption of drug on their surfaces occurred by donor–acceptor complexes and hydrogen bond formation. They were more intense in materials containing a higher concentration of functional groups. Our investigation data referring to the adsorption of paracetamol were consistent with the model of Langmuir isotherm, indicating that the drug molecules form homogeneous monolayer coverage on the surface of the carbon carriers. The pseudo-second-order model exhibited the best correlation to the kinetic results. It was shown that the amount and rate of drug release were influenced by the porosity of the materials and the total number of surface functionalities. The difference between the drug release from modified and non-modified nanocarriers was mainly because of the various acid–base properties of materials. Among all the samples tested in this study, the best material for paracetamol loading and release is C_SBA-15_-APS. This ordered mesoporous carbon exhibited optimal drug loading capacity. Moreover, a high amount of paracetamol was released within 1 h of the analysis, which was desirable. All paracetamol delivery systems based on oxidized mesoporous carbon materials exhibited high permeability through the artificial membrane.

Our future plan is to design new carbon materials decorated with other functional groups. It is assumed that, because of the introduction of different modifications onto the surface of nanomaterials, the release process may be modulated in order to obtain either sustained or immediate diffusion of paracetamol. Owing to the introduction of functional groups, the drug could be liberated in the specific site in the human body.

## Figures and Tables

**Figure 1 materials-13-04151-f001:**
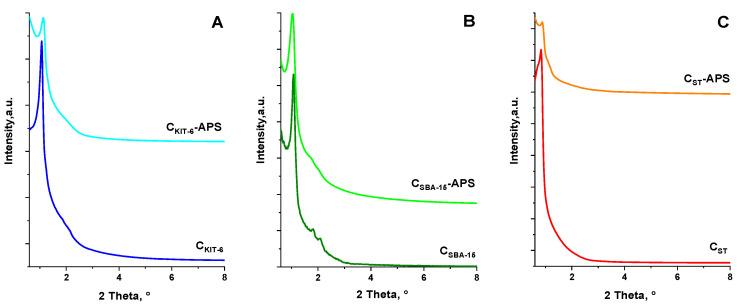
Small-angle X-ray diffraction (XRD) patterns of pristine and oxidized ordered mesoporous carbons (OMCs), (**a**) C_KIT-6_ and C_KIT-6_-APS, (**b**) C_SBA-15_ and C_SBA-15_-APS, (**c**) C_ST_ and C_ST_-APS.

**Figure 2 materials-13-04151-f002:**
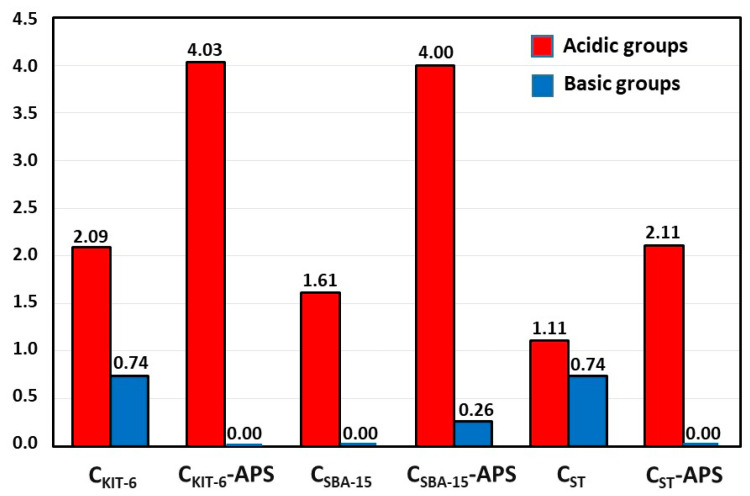
Acid–base properties of pristine and oxidized OMCs.

**Figure 3 materials-13-04151-f003:**
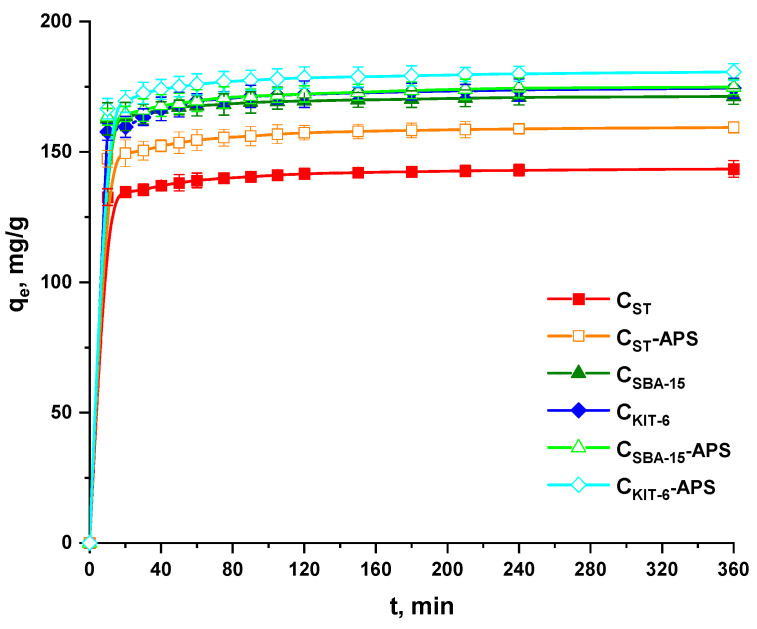
Amount of paracetamol adsorbed on the surface of OMCs as a function of contact time (initial solution concentration of drug—75 mg/L).

**Figure 4 materials-13-04151-f004:**
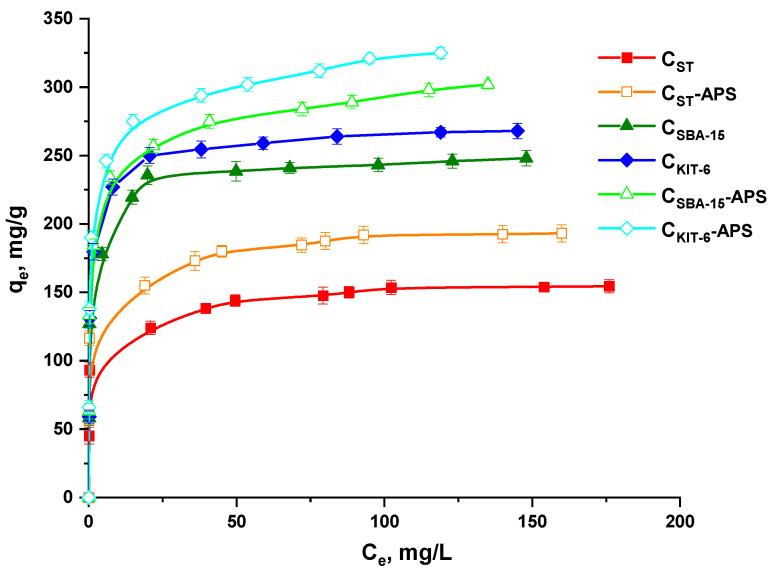
Adsorption isotherms of paracetamol onto pristine and oxidized OMCs.

**Figure 5 materials-13-04151-f005:**
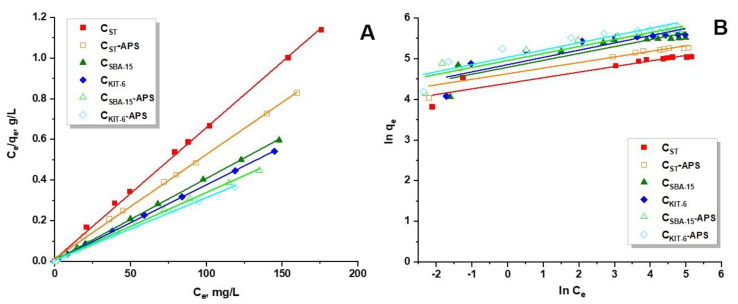
The fit of experimental data concerning adsorption of paracetamol onto pristine and oxidized OMCs to the (**a**) Langmuir and (**b**) Freundlich models.

**Figure 6 materials-13-04151-f006:**
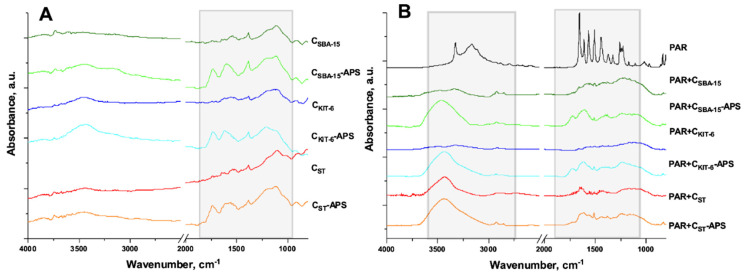
FT-IR spectra of C_SBA-15_, C_SBA-15_-APS, C_KIT-6,_ C_KIT-6_-APS, C_ST_, and C_ST_-APS (**a**) before and (**b**) after adsorption of paracetamol (PAR).

**Figure 7 materials-13-04151-f007:**
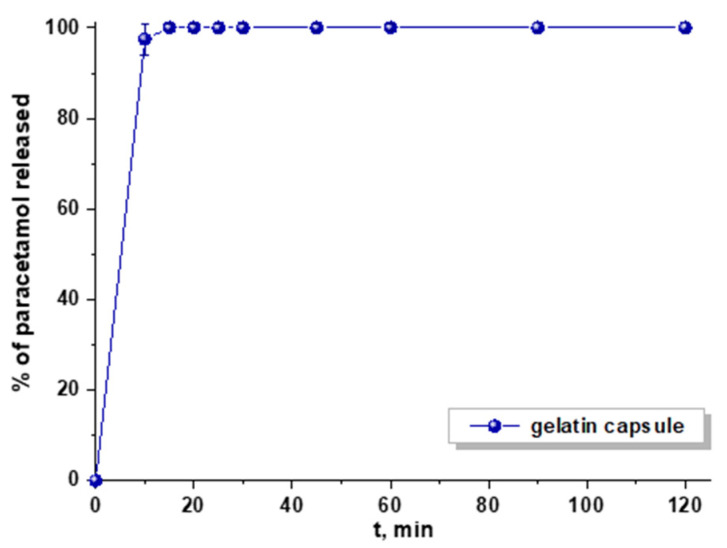
Dissolution profile of pure paracetamol from a gelatine capsule.

**Figure 8 materials-13-04151-f008:**
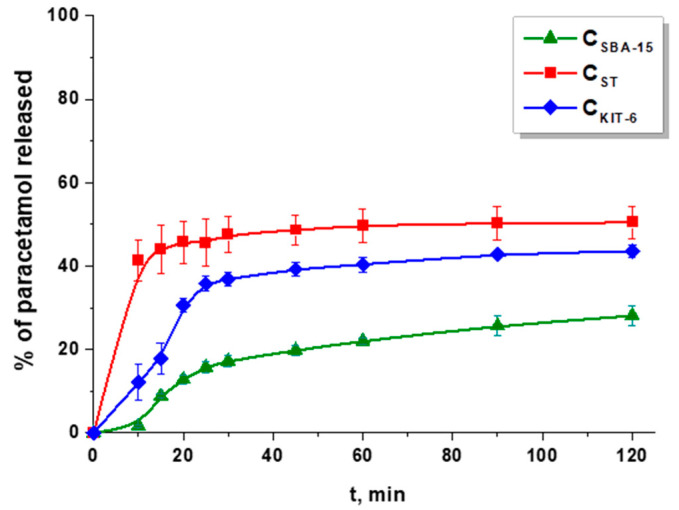
Release profiles of paracetamol from pristine OMCs.

**Figure 9 materials-13-04151-f009:**
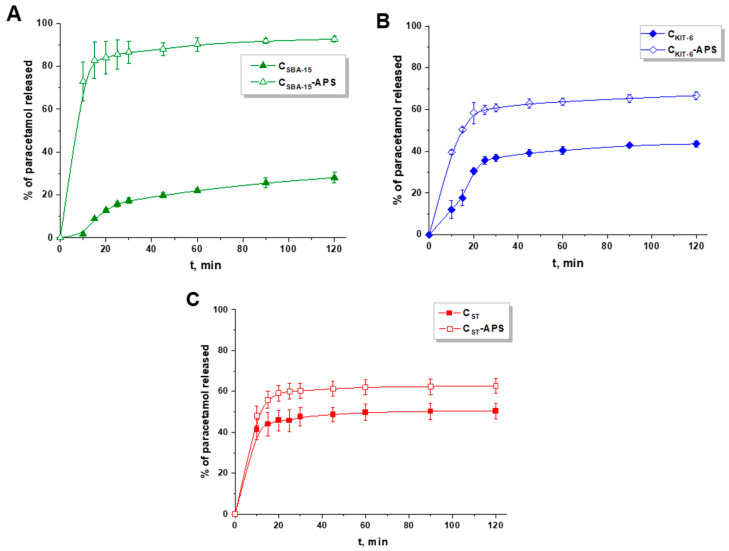
Release profiles of paracetamol from pristine and oxidized OMCs: (**A**) C_SBA-15_ and C_SBA-15_-APS, (**B**) C_KIT-6_ and C_KIT-6_-APS, and (**C**) C_ST_ and C_ST_-APS.

**Figure 10 materials-13-04151-f010:**
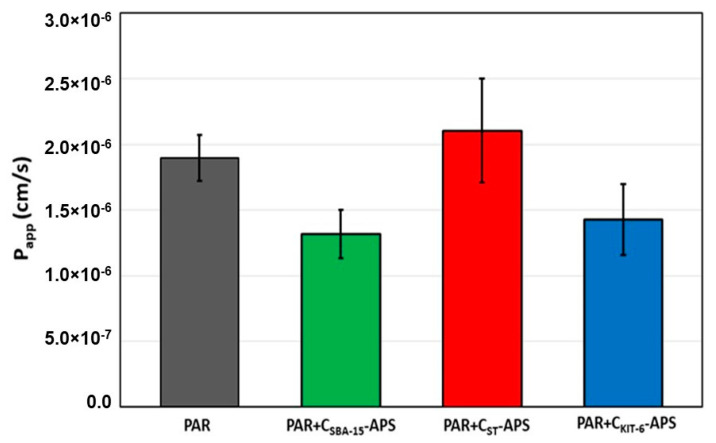
Gastrointestinal mean permeability of paracetamol from oxidized mesoporous carbons using the parallel artificial membrane permeability assay (PAMPA).

**Table 1 materials-13-04151-t001:** Interpretation of f_1_ and f_2_ factors.

	f_1_ (Difference Factor)	f_2_ (Similarity Factor)
Identical profiles	0	100
Similar profiles	0–15	50–100
Different profiles	>15	<50

**Table 2 materials-13-04151-t002:** Textural parameters of pristine and oxidized ordered mesoporous carbons (OMCs). BET, Brunauer–Emmett–Teller; APS, ammonium persulfate solution.

Material	BET Surface Area (m^2^/g)	Total Pore Volume (cm^3^/g)	Average Pore Diameter (nm)	Micropores Surface Area (m^2^/g)	Micropore Volume (cm^3^/g)
C_KIT-6_	1003	1.15	5.78	306	0.34
C_KIT-6_-APS	656	0.89	5.49	252	0.26
C_SBA-15_	986	1.47	6.54	545	0.61
C_SBA-15_-APS	689	0.95	5.49	380	0.44
C_ST_	526	0.49	4.12	231	0.14
C_ST_-APS	248	0.29	4.67	178	0.12

**Table 3 materials-13-04151-t003:** Pseudo-first-order and pseudo-second-order kinetic model parameters.

Material	q_e(exp)_ (mg/g)	Pseudo-First-Order Model			Pseudo-Second-Order Model		
		q_e(cal)_ (mg/g)	k_1_ (min^−1^)	R^2^	q_e(cal)_(mg/g)	k_2_ (g/mg·min)	R^2^
C_ST_	143	6.18	0.016	0.942	144	0.003	0.999
C_ST_-APS	159	9.94	0.019	0.939	161	0.003	0.999
C_SBA15_	171	7.35	0.018	0.927	172	0.004	0.999
C_SBA-15_-APS	174	11.93	0.022	0.925	175	0.003	0.999
C_KIT-6_	174	11.44	0.026	0.987	175	0.003	0.999
C_KIT-6_-APS	181	10.72	0.023	0.969	182	0.003	0.999

**Table 4 materials-13-04151-t004:** The parameters calculated from fitting the results of the adsorption isotherms of paracetamol onto pristine and oxidized carbon carriers to the Langmuir and Freundlich models.

Material	Langmuir			Freundlich		
	q_m_ (mg/g)	K_L_ (L/mg)	R^2^	K_F_ (mg/g (L/mg)^1/n^)	1/n	R^2^
C_ST_	154	0.602	0.999	81	0.137	0.856
C_ST_-APS	196	0.369	0.999	102	0.137	0.856
C_SBA15_	250	0.727	0.999	120	0.170	0.799
C_SBA-15_-APS	303	0.493	0.999	142	0.172	0.861
C_KIT-6_	270	0.787	0.999	128	0.178	0.801
C_KIT-6_-APS	323	0.574	0.999	153	0.178	0.860

**Table 5 materials-13-04151-t005:** Kinetics models applied to describe the release mechanism from pristine and oxidized OMCs.

Material	Zero Order	First Order	Higuchi	Hixson–Crowell	Korsmeyer–Peppas	n
C_SBA-15_	0.935	0.951	0.989	0.946	0.939	0.972
C_SBA-15_-APS	0.800	0.899	0.895	0.871	0.955	0.075
C_KIT-6_	0.667	0.647	0.969	0.629	0.737	0.197
C_KIT-6_-APS	0.864	0.913	0.936	0.880	0.953	0.510
C_ST_	0.953	0.960	0.976	0.958	0.955	0.069
Cs_T_-APS	0.734	0.766	0.860	0.756	0.948	0.096

**Table 6 materials-13-04151-t006:** The two-factor values for paracetamol (PAR) systems based on mesoporous carbon materials.

Systems	*f* _1_	*f* _2_
PAR + C_KIT-6_	72.8	7.2
PAR + C_KIT-6_-APS	70.3	12.7
PAR + C_SBA-15_	86.6	3.5
PAR + C_SBA-15_-APS	25.4	28.3
PAR + C_ST_	55.9	12.9
PAR + Cs_T_-APS	47.9	16.5

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
