# Peer review of "Design of Paracetamol Delivery Systems Based on Functionalized Ordered Mesoporous Carbons"

_materials, 2020, doi:10.3390/ma13184151_

Round 1
Reviewer 1 Report
This paper reported the design of paracetamol delivery systems. The system was based on functionalized ordered mesoporous carbons. XRD, FTIR and other methods were used for characterization of mesoporous carbons structures. In my opinion, the manuscript is interesting for readers. However, many experiments were performed with three different samples, but information provided in conclusions and abstract is too general and summarizing general trends. In my opinion, it will be useful and interesting give more information about differences in results of cubic and hexagonal structures. The manuscript can be considered for publication after minor corrections that should be revised by the authors.
Specific comments:
- Numbers should be written using dots in figure 2.
- It will be better to provide results with standard deviation. For instance, data in Table 2.
- Scientists often use paracetamol for evaluation of delivery systems. It will be better to compare results with other papers.
Reviewer 2 Report
The authors synthesized multiple mesoporous carbon and oxidized the materials with acidic solution to increase drug loading efficiency. Then the releasing profiles of these materials were compared to gelatine capsules. The authors did a great job in data analysis, however, critical discussion of the results are missing. Moreover, the paper is lack of novelty, the authors should highlight the novelty of the work more in the abstract and introduction. Figure 2. showed basic group decreased after APS treatment in CKIT and CST, but increased in CSBA-15. Could the authors provide some explanation? Figure 3. How many replicates did authors do? Is there a statistical difference between CSBA15, CSBA15-APS and CKIT? Figure 6 is blurry. Please provide a figure with higher resolution. The authors should provide FT-IR of mesoporous carbon without PAR for comparison. The current FT-IR of mesoporous carbon loaded with PAR does not provide any valuable information. Figure 8, 9. What is your target releasing time frame? Using mesoporous carbon only extended the maximum release from 10 minutes to 20 minutes. Is it a significant difference that could prevent overdose? Also, why is there a significant difference in CKIT-APS and CKITSBA-15? Is it because of the mesoporous structure or pore size? Discussion is missing here. Among the materials you tested, which one is the best for drug loading and releasing? Any suggestions of material design for drug loading and control releasing?
Round 2
Reviewer 2 Report
The authors have addressed my comments.